# AN EFFECTIVE GCN-BASED HIERARCHICAL MULTI-LABEL CLASSIFICATION FOR PROTEIN FUNCTION PREDICTION

## ABSTRACT

We propose an effective method to improve Protein Function Prediction (PFP) utilizing hierarchical features of Gene Ontology (GO) terms. Our method consists of a language model for encoding the protein sequence and a Graph Convolutional Network (GCN) for representing Go terms. To reflect the hierarchical structure of GO to GCN, we employ node(GO term)-wise representations containing the whole hierarchical information. Our algorithm shows effectiveness in a large-scale graph by expanding the GO graph compared to previous models. Experimental results show that our method outperformed state-of-the-art PFP approaches.

## 1 INTRODUCTION

Protein Function Prediction (PFP) is one of the key challenges in the post-genomic era (Zhou et al., 2019; Li et al., 2018). With large numbers of genomes being sequenced every year, the number of novel proteins being discovered is expanding as well (Spalević et al., 2020). On the other side, protein functions are reliably determined in wet-lab experiments which are cumbersome and high-cost. As a result, the number of novel protein sequences without function annotations is rapidly expanding. In fact, UniRef100 (Consortium, 2019) contains over 220M (million) protein sequences of which less than 1M have function annotations proved by experiments (Littmann et al., 2021). Fast and accurate PFP is especially important in biomedical and pharmaceutical applications which are associated with specific protein functions.

Protein functions are defined by Gene Ontology (GO) composed of a directed acyclic graph (DAG) (Ashburner et al., 2000). There are three GO domains: Molecular Function Ontology (MFO), Biological Process Ontology (BPO), and Cellular Component Ontology (CCO), where each node represents one function called GO term, and each edge represents a hierarchical relation between two GO terms, such as 'is_a', 'part_of', and etc. Since one protein is usually represented by multiple function annotations, PFP can be regarded as hierarchical multi-label classification (HMC).

There are two groups of existing approaches for PFP: local approaches and global approaches. Local approaches usually constructed a classifier for each label (GO term) or for a few labels of the same hierarchy level (Lobley et al., 2008; Minneci et al., 2013; Cozzetto et al., 2016; Rifaioglu et al., 2019). On the other hand, global approaches constructed a single classifier for multiple labels. The initial global approaches considered PFP to flat multi-label classification, ignoring the hierarchical structure of GO, and considering each label independently (Kulmanov & Hoehndorf, 2020). Recent global approaches have constructed a structure encoder to learn the correlation among labels (Zhou et al., 2020; Cao & Shen, 2021). However, these results showed that existing global approach-based models had limitations in representing correlations between GO terms by learning a large-scale hierarchical graph of GO.

One of the structure encoders in global approach-based models used Graph Convolutional Network (GCN). GCN has been applied in learning representation of node features. Nevertheless, in the case of applying GCN to a large-scale hierarchical graph, it was difficult to obtain information among long-distance (Chen et al., 2019; Zeng et al., 2021) and unable to obtain adequate structure information since adjacent nodes did not contain any hierarchical features (Hu et al., 2019). To overcome these shortcomings, we build node-wise representations containing the whole hierarchical information, which is involved relationship between long-distance nodes and structure information.

In this paper, we propose a novel PFP model that combines a pre-trained Language Model (LM) and GCN-based model including new node-wise representations. A pre-trained LM as a sequence encoder (Littmann et al., 2021) extracts general ad helpful sequence features. GCN-based model as a structure encoder blends hierarchical information into a graph representation to improve GCN performance in a large-scale hierarchical graph of GO. To predict the probability of each GO term representing target protein functions, the prediction layer is constructed as a dot product of outputs of two encoders. The experimental results show that our method achieves performance improvement compared to the-state-of-the art models, especially in the most difficult BPO.

## 2   RELATED WORK

### 2.1   PROTEIN SEQUENCE FEATURE EXTRACTION

Protein sequences contain multiple biophysical features related to function and structure. Initially, protein biophysical features such as motifs, sequence profiles, and secondary structures were calculated from a suite of programs and then utilized as protein sequence feature vectors by combining them (Lobley et al., 2008; Minneci et al., 2013; Cozzetto et al., 2016; Rifaioglu et al., 2019). While these methods intuitively utilized the relationship between protein features and its biological functions, it required deep knowledge of proteomics and had high-cost.

Various deep learning architectures that extract high-level biophysical features of protein sequences have been proposed. Convolutional Neural Network (CNN) is one of the architectures as a sequence encoder to learn sequence patterns or motifs that are related to functions (Xu et al., 2020). Therefore, 1D CNN was utilized as effective sequence encoder in previous researches (Kulmanov & Hoehndorf, 2020; Zhou et al., 2020; Kulmanov et al., 2018).

With the advent of transformers (Vaswani et al., 2017), which is attention-based model, in Natural Language Processing (NLP), various attention-based LMs were applied to protein sequence embedding (Rao et al., 2019; Vig et al., 2020; Rives et al., 2021; Heinzinger et al., 2019; Elnaggar et al., 2020). As protein sequences can be considered as sentences, these learned the relationship between amino acids constituting the sequence and learned contextual information. SeqVec (Heinzinger et al., 2019) and ProtBert (Elnaggar et al., 2020), which were learned protein sequences using ElMo (Peters et al., 2018) and BERT (Devlin et al., 2018), showed that these mostly extracted biophysical features of protein structures and functions, such as secondary structures, binding sites, and homology detections.

### 2.2   PROTEIN FUNCTION PREDICTION (PFP)

PFP methods can be categorized into two different approaches which are local and global. Local approaches commonly employed single or few multi-label classifiers to each or few GO terms. These approaches included FFpred (Lobley et al., 2008; Minneci et al., 2013; Cozzetto et al., 2016) and DEEPred (Rifaioglu et al., 2019). FFpred predicted one GO term by multiple Support Vector Machines (SVMs) trained with radial basis function kernels to recognize protein sequence patterns associated with the GO term (Cozzetto et al., 2016). DEEPred created a multi-label classification model using deep neural network for each GO hierarchical level (Rifaioglu et al., 2019). Each model could carry out five GO terms in most labels. Even though this procedure generated 1,101 different models concerning all GO domains, this still needed numerous models for PFP (Rifaioglu et al., 2019). Ultimately, local approaches required expensive costs by training numerous models.

On the other hand, global approaches respectively constructed one classifier model for MFO, BPO, and CCO. The initial global approaches considered PFP as flat multi-label classification. They focused on extracting function-related features from the protein sequence. One of the function-related features is motifs, called sequence patterns. DeepGoPlus (Kulmanov & Hoehndorf, 2020) encoded sequence to extract motifs using 1D CNN and then predicted the probability of annotating each GO term using one fully connected layer. Compared to local approach-based previous methods, DeepGoPlus achieved improved performance in PFP, despite its simple model architecture. This model resulted in poor performance when the number of GO terms increases. The recent global approaches expanded that built a structure encoder to improve performance. In DeepGOA (Zhou et al., 2020), all GO terms were regarded as correlated labels contrary to DeepGoPlus where they were regarded

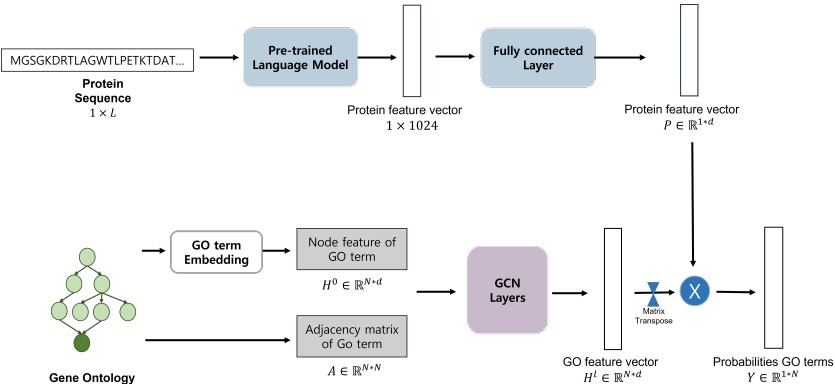

Figure 1: The network architecture of this work. we defined that $L$ is the protein sequence length, $d$ is the hidden dimension size and $N$ is the number of GO terms in each GO domain.

as independent labels. This enabled GCN to learn more effectively GO terms. They showed improved performance compared to DeepGoPlus, although the same protein sequence encoding was used. They extracted patterns using 1D CNN, but could not extract various features related to functions other than patterns. TALE (Cao & Shen, 2021) implied transformer encoder (Vaswani et al., 2017) for sequence encoder and embedded GO with hierarchical information of each GO term. They learned various and specific features related to function as the transformer encoder. However, this indirectly learned the correlation among GO terms since they did not learn GO itself and just use hierarchical information of each GO term.

One of the common problems of global approaches is that they partially used GO terms. In general, a cut-off criterion in each GO domain was a number of annotations such as 25, 50, and 150. Global approaches, which had the cut-off criterion, utilized about 13% of the total GO terms. Although TALE (Cao & Shen, 2021), which is the latest model, had the cut-off criterion to 1, this still utilized only about 60% GO terms for their model. In this paper, we propose a method that effectively learns the relation between GO terms in an extended data using over 85% GO terms. This model results in similar or higher performance to the latest models used fewer GO terms.

## 3    PROPOSED METHOD

In this section, we describe the details of our model. The overall architecture is shown in Figure 1. This model has two inputs: a protein sequence and the hierarchical graph of GO. A protein sequence is encoded by pre-trained LM and is reduced dimension to the feature vector size of GO respectively. A GO graph is represented to a large-scale adjacency matrix and node-wise feature matrix. These matrices are inputs to GCN for learning the hierarchical representation of GO terms. The prediction layer is built as a dot product of a protein feature vector and a GO terms vector to predict the probability of annotated GO term of each protein sequence. We explain the technicality of these process in this subsection.

### 3.1    PROTEIN SEQUENCE ENCODING

We employ pre-trained LM as a sequence encoder. As we mentioned in related work, pre-trained LM, such as SeqVec (Heinzinger et al., 2019) and ProtBert (Elnaggar et al., 2020), already proved their performance to capture rudimentary features of proteins such as secondary structures, biological activities, and functions (Rives et al., 2021; Vig et al., 2020). Especially, it was showed that SeqVec (Heinzinger et al., 2019) is better than ProtBert (Elnaggar et al., 2020) to extract high-level features related functions for PFP (Littmann et al., 2021). Seqvec (Heinzinger et al., 2019) is utilized as a protein sequence encoder. This makes the various lengths of protein sequences to $1 \times 1024$ representation vectors with high-level biophysical features. Protein sequence representations are converted to $P \in \mathbb{R}^{1 \times d}$ low-dimensional representation vectors by fully connected layer to combine GO term vectors.

### 3.2 HIERARCHICAL REPRESENTATION OF GO TERM

Initial node features $H^0 \in \mathbb{R}^{N \times N}$ are represented as a one-hot encoding matrix where the $i$ th row GO term and its ancestors are 1. The GO term Embedding layer reduces dimension by converting sparse matrix to $H^0 \in \mathbb{R}^{N \times d_0}$ dense matrix for preventing overfitting and reducing training time in GCN. It indicate that node features contain its physical location and conceptual information in a hierarchical graph. $N$ is a number of GO terms(node) and $d_0$ is a scale of dimension, which is the maximum depth in each domain for containing hierarchical information in the dense matrix.

The adjacency matrix $A \in \mathbb{R}^{N \times N}$ contains the relationship between GO terms. When a parent node is $t$ and the children node is $s$, the adjacency matrix is combined prior probability $P(U_s|U_t)$ with Information Content(IC) (Song et al., 2013) that measures semantic similarity between $t$ and $s$. The existing adjacency matrix is usually built by one-hot encoding or the prior probability. The one-hot encoding can not involve any additional information other than connection information between GO terms. The prior probability can involve relational information. Nevertheless, it apply the inveterate label imbalanced problem in the PFP dataset to the adjacency matrix due to being highly dependent on the training dataset. We solve this problem that adding IC, which is less affected by the training dataset (Zhou et al., 2020). The adjacency matrix is defined as follows:

$$A = P(U_s|U_t) + \frac{IC(s)}{\sum_{i \in child(t)} IC(i)} \tag{1}$$

Prior probability $P(U_s|U_t)$ is calculated as follows:

$$P(U_s|U_t) = \frac{P(U_s \bigcap U_t)}{P(U_t)} = \frac{P(U_s)}{P(U_t)} = \frac{N_s}{N_t} \tag{2}$$

Where $U_t$ means a number of annotations in the training dataset, $P(U_s|U_t)$ means the conditional probability that $t$ and $s$ are annotated in the same protein sequence. IC is calculated as follows:

$$IC(k) = -\mathrm{log}p(k)$$

$$p(k) = \frac{freq(k)}{freq(root)} \tag{3}$$

$$freq(k) = U_k + \sum_{i \in child(k)} freq(i)$$

Where $p(k)$ is probability of each GO term $k$ in the GO dataset, $freq(k)$ is frequency of $t$ and $child(k)$ is every children of $k$.

Initial node features $H^0$ are updated $H^l \in \mathbb{R}^{N \times d}$ with node features of adjacent nodes through lth GCN layer ($1 \leq l \leq M$). GCN layer is represented as follows:

$$H^{l+1} = ReLU(\hat{A}H^l W^l) \tag{4}$$

### 3.3 PREDICTION LAYER AND LOSS FUNCTION

The prediction layer builds a dot product of protein sequence feature vectors and GO feature vectors to finally predict the probability of each GO term for the one protein sequence. We use the sigmoid function in the prediction layer. The prediction layer is defined as follows:

$$Y = sigmoid(H^T P) \tag{5}$$

We use binary-cross entropy as the loss function since it is binary problem at each GO term.

## 4 EXPERIMENTS

### 4.1 DATASET

The CAFA3(Zhou et al., 2019) dataset, which was used in an international protein function prediction competition, was used for our experiment. According to CAFA3, experimental annotations have one of 8 experimental evidence codes: EXP, IDA, IPI, IMP, IGI, IEP, TAS, and IC. The training dataset includes protein sequences with all known experimental annotations before September 2016. The test dataset includes protein sequences with their experimental annotations known between September 2016 and November 2017. The GO dataset is a version of May 31, 2016. This GO dataset is also used in CAFA3. GO dataset has 11,888 GO terms in molecular function(MFO), 30,546 GO terms in biological process(BPO) and 4241 GO terms in cellular component(CCO).

We propagated annotations using the true path rule. For example, if a protein sequence 'P' has annotation 'A' and GO term 'B' is an ancestor of 'A', protein sequence 'P' contains annotations both 'A' and 'B'. Among various type relations, such as 'is_a', 'part_of', and etc, we propagated with only an 'is_a' type relation as ancestor GO term, as same as a CAFA3 assessment tool. We constructed hierarchical graph of GO as input excepting isolated GO term. Eventually, we utilized over 85% of all GO terms in MFO, over 90% of all GO terms in BPO and CCO. As a result, our model utilized about 15% larger GO graph than previous models.

| Dataset | Statistics | MFO | BPO | CCO |
|---|---|---|---|---|
| | Seq in Training Set | 35,086 | 50,813 | 49,328 |
| CAFA3 | Seq in Test Set | 1,101 | 2,145 | 1,097 |
| | Number of GO terms | 10,236 | 28,678 | 3,905 |

Table 1: Statistics of sequences and GO terms in CAFA3

### 4.2 PERFORMANCE ON THE CAFA3

**Experimental settings** We constructed each GO domain's model since each domain (MFO, BPO, CCO) had a different number of GO terms. Sequence and GO term's hidden dimensions $d$ were each GO graph's maximum depth, which was 41 dimensions of MFO, 80 dimensions of BPO, and 54 dimensions of CCO. Although the maximum depth of BPO was 155 since there were only 42 GO terms with a depth of 80 or more, we used a maximum depth of BPO to 80 for efficient training time.

**Evaluation** We evaluated our model using maximum protein-centric F-measure ($Fmax$) and area under the precision-recall curve (AUPR). $Fmax$ is the official assessment score used in CAFA3. $Fmax$ is defined as follows:

$$Fmax = \max_t \left( \frac{2\overline{pr(t)} \cdot \overline{rc(t)}}{\overline{pr(t)} + \overline{rc(t)}} \right) \tag{6}$$

Where $\overline{pr(t)}$ and $\overline{rc(t)}$ is average precision and recall in threshold $t$. $\overline{pr(t)}$ and $\overline{rc(t)}$ are calculated as follows:

$$\overline{pr(t)} = \frac{1}{G(t)} \sum_{i=1}^{G(t)} \frac{tr_i \cdot \widehat{tr_i}(t)}{|tr_i|_1}$$

$$\overline{rc(t)} = \frac{1}{n} \sum_{i=1}^{n} \frac{y_i \cdot \widehat{tr_i}(t)}{|tr_i|_1} \tag{7}$$

Where $G(t)$ is the number of proteins with at least one GO term in our test dataset, $n$ is the number of all proteins in the test dataset. $tr_i$ is i-th GO term's true vector. $|tr_i|_1$ is that if the i-th GO term is true, $tr_i$ is 1; otherwise zero. $\widehat{tr_i}(t)$ i-th GO term's predicted vector at threshold $t$. Thresholds $t \in [0, 1]$ increase with 0.01 steps.

On the other hand, AUPR is used to evaluate classification problems, especially highly imbalanced-label classification. We used AuPRC on each GO term to evaluate the performance.

| Method | $Fmax$ | | | AUPR | | |
|---|---|---|---|---|---|---|
| | MFO | BPO | CCO | MFO | BPO | CCO |
| Naive | 0.331 | 0.253 | 0.541 | 0.312 | 0.173 | 0.483 |
| DIAMONDScore | 0.532 | 0.382 | 0.523 | 0.461 | 0.304 | 0.500 |
| DeepGoCNN | 0.411 | 0.388 | 0.582 | 0.402 | 0.213 | 0.523 |
| TALE | **0.548** | 0.398 | **0.654** | 0.471 | 0.317 | 0.626 |
| Ours | 0.518 | **0.470** | 0.637 | **0.476** | **0.368** | **0.626** |

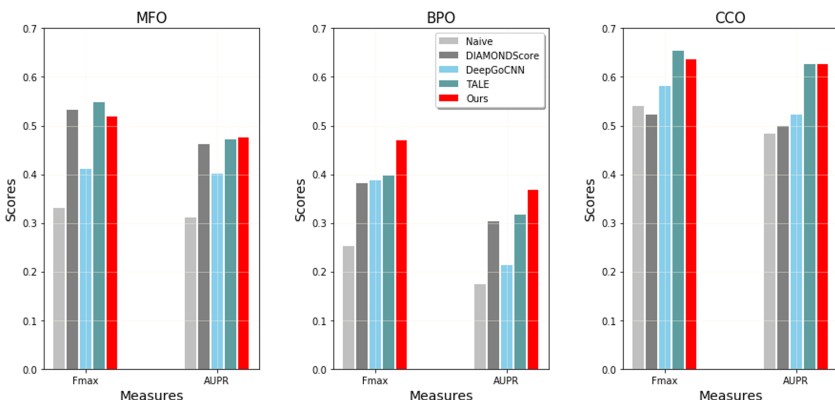

Figure 2: The performance of our model against baseline models on the CAFA3

**Baseline Models and Results** We compared our model using baseline models. The baseline models were Naive, DIAMONDScore, DeepGoCNN, and TALE. Naive predicted protein functions using prior probability. DIAMONDScore predicted protein functions based on the sequence similarity measured using the Diamond tool (Buchfink et al., 2015). DeepGoCNN was the most famous sequence-based PFP model, using 1D CNN protein sequence encoder and a flat multi-label classifier (Kulmanov & Hoehndorf, 2020). TALE was the state-of-the-art sequence-based PFP model, using a transformer encoder for protein sequences encoding and embedding GO terms with hierarchical information (Cao & Shen, 2021). Results were shown in Figure 2. Overall, our model achieved the best performance on all domains in AUPR and on biological process (BPO) in $Fmax$. Our model especially improved from 0.398 to 0.470 on BPO (by 18%) compared to TALE. All the baseline models did not exceed 0.4 in $Fmax$ on BPO, which had the most complex and large-scale hierarchical graph. It proved that our model was effective in a large-scale hierarchical graph, as it showed highly improved performance on BPO.

## 5 CONCLUSION

In this paper, we proposed an effective GCN-based model to improve the protein function prediction. We build initial node-wise representations involving the whole hierarchical information to overcome the shortcoming of GCN-based model in a large-scale graph. After that, we combined a pre-trained LM and an effective GCN-based model to predict protein function.

Experimental results showed that in $Fmax$, our model outperformed the related state-of-the-art methods on BPO, which was the highly difficult domain with the most large-scale hierarchical graph. Moreover in AUPR, our model outperformed the related state-of-the-art methods on all domains. We demonstrated that our model was especially effective in a large-scale graph having deep depth.

Learning correlation of GO terms is an important part of PFP since it is the large-scale hierarchical multi-label classification. It is necessary to extract the features of the protein sequence related to

the function from the PFP. We used a pre-trained LM in the sequence encoder part. It can properly extract the features of the protein sequence, but does not train the encoder while training the model. We will work later on sequence encoders that can extract function-related features using different LMs.

## 6    REPRODUCIBILITY STATEMENT

The learning was processed with 10 epochs and 32 batch sizes. We used ReLU activation function, 1e-3 as the learning rate, and Adam optimizer (Kingma & Ba, 2014). We simulated our model with NVIDIA TITAN Xp. The code for reproducibility is posted at the anonymous GitHub repository : `https://anonymous.4open.science/r/Reproducibility-code`

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
