# OpenReview forum: "An Effective GCN-based Hierarchical Multi-label classification for Protein Function Prediction"
_ICLR.cc/2022/Conference — ICLR 2022 Submitted_

### Official Review · Reviewer_evqx · 2021-10-28

**Correctness:** 4
**Technical Novelty And Significance:** 2
**Empirical Novelty And Significance:** 2
**Recommendation:** 3
**Confidence:** 5

**Main Review:**

##########################################################################

Pros:

- The idea of this paper is very intuitive and understandable, including a language model for encoding the protein sequence and a Graph Convolutional Network (GCN) for representing Go terms.
- Overall, the paper is well written. In particular, the INTRODUCTION section has a nice flow and puts the related works into context. Despite the method having limited novelty (LM + GCN), the method has been well-motivated by pointing out the limitations in SOTA methods.
- The EXPERIMENTS section is well structured. In partuical, the author provides the statistics of sequences and GO terms in CAFA3, the experimental settings and evaluation are also introduced in great detail.

##########################################################################

Cons:

- The core idea of this paper is the PFP model, which combines a pre-trained Language Model (LM) and GCN-based model including new node-wise representations. However, both LM and GCN are now very mature methods, and the combination of the two is limited novelty.
- The experimental part of this paper is only compared on one dataset: CAFA3. It is difficult to convince others that the proposed method is really effective.
- In the comparison between the author's method and the baseline methods, the TALE is superior to the proposed method in $F_{max}$, but the author did not give a reasonable explanation.


**Summary Of The Paper:**

This paper proposes a novel PFP model that combines a pre-trained Language Model (LM) and GCN-based model including new node-wise representations, to improve Protein Function Prediction (PFP) utilizing hierarchical features of Gene Ontology (GO) terms. The experimental results show that the proposed method achieves performance improvement compared to SOTA models, especially in the most difficult BPO.

**Summary Of The Review:**

Considering the above pros and cons, I think there is still a lot of room for improvement in this paper.

---

### Official Review · Reviewer_mvKR · 2021-10-31

**Correctness:** 2
**Technical Novelty And Significance:** 3
**Empirical Novelty And Significance:** 3
**Recommendation:** 3
**Confidence:** 5

**Main Review:**

# Overall Impression
I believe the proposed method to be interesting and the contribution to the GO embedding technique
to be valuable for the PFP communitt. However, I found the paper very difficult to read, and I have several
concerns about this study, which I detail below.

# Major Concerns

1. Evaluation is lacking representative state-of-the-art competitors.

The authors compare their method against 4 competitors:
 * Naive: a simple frequency based predictor.
 * DIAMONDScore: A sequence similarity based predictor.
 * DeepGOCNN: A Deep Learning method that uses a Convolutional Neural Network
 * TALE: A transformer-based PFP method.

Notably, the authors decided not to compare with GOLabeler, the best performing method
from the CAFA3 competition. I also find it very strange that they decided to go only with
DeepGOCNN, which is consistently outperformed by the full architecture (DeepGOPlus) in the paper
by Kulmanov and Hoehndorf (Bioinformatics, 2020).

The authors claim their method surpasses the state-of-the-art, but this is not properly shown
in my opinion. How does the proposed method compare to GOLabeler and DeepGOPlus?


2. The proposed method is not explained in the paper.

The method proposed by the authorsis not very well explained in the manuscript.
I would not have understood the architecture without careful studying of the code.

Below are some questions that would have remained unanswered had I not been willing to
carefully inspect and study the code myself:
 * $H^0$ gets compressed from a dimensionality of $N \times N$ to $N \times d_0$, but this embedding layer is never described.
 * Node features get updated through layers $l$ using equation 4. What is $\hat{A}$? What is $W^l$, perhaps the weights of this layer? If so, why is this not properly explained?
 * When explaining equation 2, what are $N_s$ and $N_t$?

3. The code is lacking instructions or documentation.

The understanding of this manuscript requires the reader to be a seasoned Python programmer with knowledge of several very specialised frameworks.
Moreover, the provided repository comes with no instruction or documentation on how to use the code to reproduce their results.
Had I not been willing to spend several hours carefully studying this code, I would consider this paper as not reproducible.

# Minor Comments

Section 2 does not represent a good summary of the field in my opinion. Based on the brief
description of the field in this section, the complexity and wide variety of existing protein
function prediction methods is not evident. Moreover, the state-of-the-art is hardly reflected
in this section, which fails to mention the top-performing methods in the CAFA competition.

The spelling of existing models contains several mistakes. Examples are "DeepGoPlus" (the correct spelling is DeepGOPlus)
and DeepGoCNN (the correct spelling is DeepGOCNN).

There are several grammatical and words that seem to be expressing the wrong idea. For example, in Section 2.2, the authors state
"TALE (Cao & Shen, 2021) implied transformer encoder [...] for a sequence encoder and embedded GO with hierarchical information of each GO term."
It seems to me that "implied" should be replaced by "applied", to describe the work by Cao & Shen more accurately. Several other instances exist in
the text.

**Summary Of The Paper:**

The authors introduce a GCN-based hierarchical model for protein function prediction.
The idea is built on top of using existing protein embedding techniques and using a GCN
to embed GO terms and their hierarchical information, encoded as node features.

**Summary Of The Review:**

This article proposes an interesting architecture for PFP, but the evaluation is not very complete, and the explanation of the method is very confusing.

---

### Official Review · Reviewer_BnVc · 2021-11-01

**Correctness:** 2
**Technical Novelty And Significance:** 1
**Empirical Novelty And Significance:** 1
**Recommendation:** 3
**Confidence:** 4

**Main Review:**

Strength: The paper is easy to read.

Weakness: Overall the proposed method is very similar to DeepGOA, which the paper cited but did not benchmark with. The main difference is that DeepGOA used a CNN model trained from scratch to model protein sequences while here a pretrained language model was used.

Comment: Some ablation study on different components of the model would be helpful. Particularly the sequence embedding part of the model. Better protein language models e.g. ProtBert (cited), ESM may help the performance.

**Summary Of The Paper:**

The paper proposes a method to predict protein functions from Gene Ontology (GO) and protein sequences. The protein sequences are embedded with a pretrained protein language model (SeqVec) and the GO network is modelled with a graph convolutional neural network. The method was benchmarked using CAFA3 competition datasets. Improved model performance was shown against a Naive baseline, DIAMONDScore, DeepGoCNN, and TALE.


**Summary Of The Review:**

I find the paper not particularly novel. The paper needs more discussion and comparison with DeepGOA. The method itself is not novel enough for ICLR.

---

### Official Review · Reviewer_k9w6 · 2021-11-02

**Correctness:** 2
**Technical Novelty And Significance:** 2
**Empirical Novelty And Significance:** 2
**Recommendation:** 1
**Confidence:** 5

**Main Review:**

Strengths:
- Model uses more of the GO terms than other work that trim infrequent terms
- Model achieves better performance on BPO term annotation prediction

Weaknesses:
- Limited novelty as it combines two existing models, closely following the approach of [1] but replacing the CNN-based sequence embedding with [2].
- Discussion of related work and empirical analysis is lacking as not all appropriate baselines are considered and novelty vs existing work is not properly described/motivated.

Detailed comments:
Approach:
I believe just after equation (2) where the text says that “U_t means a number of annotations in the training dataset,” this should be N_t. Otherwise, what are N_t and N_s? These are not defined anywhere. Likewise, equation (3) should be freq(k) = N_k + \sum_ …. This is the same adjacency weighting scheme used by DeepGOA [1]. DeepGOA was not included in the experiments to validate the choice of sequence embedding, this method’s main difference with DeepGOA.

Regarding how many GO terms to include in the model: An ablation study could be included to justify using “85% of all terms” vs the 13% (according to this paper) used in previous work. It is not well justified to include more terms without an experiment to show the advantage of this design decision. More terms may indeed cause overfitting and other issues, eg increasing the path length between correlated/useful GCN nodes. As this seems to be a main contribution of this work, such a study would help strengthen the contribution.

Related works:
The related works section, especially the last two paragraphs of 2.2 which discuss the most closely-related works is unclear. It does not clearly position the current work in context of these other deep structured PFP models and it fails to properly motivate the work by clearly describing shortcomings or limitations of these past works. This is not to say that these works do not have limitations, just that this text does not make it clear what gaps it is filling/addressing. In particular, the differences with DeepGOA [1] should be clearly discussed.

“Cut off-criterion” is not well explained. Is this referring to the number of times a GO term had to be seen in a dataset in order to include it in the GO DAG? If so, please make sure to clarify this point.

A closely related PFP work that I suggest be included in related work is:
Li J, Wang L, Zhang X, Liu B, Wang Y. 2020. Gonet: a deep network to annotate proteins
via recurrent convolution networks. In: 2020 IEEE international conference on
bioinformatics and biomedicine (BIBM), volume 2. Piscataway: IEEE, 29–34

Mostly minor comments on clarity and text:
Several typos and language ambiguities should be addressed.
-Abstract: “Go terms” should be “GO terms”
-Section 1
“On the other side,” not clear what this means. On the other side of what?
“Protein functions are defined by Gene Ontology (GO) composed of a …” -> “Protein functions are *described* by the Gene Ontology (GO) composed of a …”
“GCN has been applied…” -> “GCNs have been applied…”
“...which is involved relationship between…” What does this mean? This language is unclear. Is this saying the hierarchical information is *encoding* relationships between nodes at long distances? That it is *capturing* that relationship? Or something else?
“Extracts general ad helpful” -> “Extracts general *and* helpful”
-Section 2
“it required deep knowledge of proteomics and had high-cost” -> “*they* required deep knowledge of proteomics and had *high cost*.” Also, what is meant by “high cost” here?
“...(CNN) is one of the architectures as a sequence…”->“...(CNN) is one of the architectures *used* as a sequence…”
“transformers…,which is attention-based model” ->“transformers…,which *are* attention-based models”
“The recent global approaches expanded that built a structure encoder…” I am not sure what this means
“TALE implied …”-> “TALE employed …”
-Section 3
“A GO graph is represented to” -> “A GO graph is represented *by*”
What does “large-scale adjacency matrix” mean in this context?
“We explain the technicality of these process in this subsection” -> “We explain the *technical details* of *this* process in *the following* subsection”
“It was showed that” -> “It was shown that”
“...to extract high-level features related functions for PFP.” This is not clear. To what functions does this sentence refer, functions of the features? Functions of the protein that are to be learned? It is not clear.
“Seqvec” -> “SeqVec”
“The adjacency matrix is combined prior probability”->“The adjacency matrix combines prior probability...”
“Nevertheless, it apply the inveterate…” It is not clear what “applies” and what it applies. Is this the wrong word here or what is the sentence saying?
“label imbalanced problem”->“label imbalance problem”
-Section 4
“The GO dataset is a version of May 31, 2016”->“The GO dataset is a version *from* May 31, 2016”?
“as same as a CAFA3 assessment tool” ->“same as a CAFA3 assessment tool”
“GO as input excepting isolated”->“GO as input excluding isolated”


**Summary Of The Paper:**

This paper presents a model to predict Gene Ontology (GO) term annotations for protein function. The model uses an existing method, SeqVec [2], to encode the protein sequence and a GCN on the Gene Ontology (GO) DAGs to encode the structure of term relationships. Like in DeepGOA[1], the graph is weighted by functions of term frequencies. Sequence embedding is reduced via FC layers to dimension d, where d is equal to the DAG depth for the ontology being predicted. The final prediction is a dot product between the GCN encoding and the reduced sequence embedding. The model largely combines these two existing models.

[1] Zhou, Guangjie, et al. "Predicting functions of maize proteins using graph convolutional network." BMC bioinformatics 21.16 (2020): 1-16.
[2] Heinzinger, Michael, et al. "Modeling aspects of the language of life through transfer-learning protein sequences." BMC bioinformatics 20.1 (2019): 1-17.
[3] Cao, Yue, and Yang Shen. "TALE: Transformer-based protein function Annotation with joint sequence–Label Embedding." Bioinformatics 37.18 (2021): 2825-2833.

**Summary Of The Review:**

Technical contributions are very limited since the GO embedding module uses the same structure weighting, architecture, and loss function as existing method DeepGOA [1] and replaces its sequence embedding module with an existing protein sequence encoder, SeqVec [2]. Empirical results do not significantly improve, or in some cases even match, the previous method TALE [3], with the exception of prediction on the BPO.

The main difference is in the dataset in that it uses all GO terms that have a `is_a’ relation in the DAG, rather than the common approach of removing terms that are not seen or seldom seen in training. However later in the text it states that more deep BPO terms were also trimmed when the maximum depth was cut off to 80, so it is not clear which terms are used to evaluate after all.

Motivation and comparison to prior work is very limited as the paper does not properly discuss differences to prior work or motivate the decision made. In particular, the choice of term “cut off” (which terms are used in training) are not empirically validated; the use of these particular architectures are not well-supported as the component modules are not used as baselines in experiments.

---

### Decision · Program_Chairs · 2022-01-20

**Decision:**

Reject

**Comment:**

The paper proposes a method to predict protein functions from Gene Ontology (GO) and protein sequences. The protein sequences are embedded with a pretrained protein language model (SeqVec) and the GO network is modelled with a graph convolutional neural network.

Reviewers found the paper well-written and structured. At the same time, they found the novelty of the paper limited. Two reviewers pointed out that the paper is very similar to DeepGOA, which the authors cite but don't compare against. Overall, there is consensus among the reviewers that the paper is not suitable for ICLR.

The authors didn't submit a rebuttal.

We encourage the authors to take into account reviewer comments to improve the paper. Since it is more on the application side, perhaps a computational biology conference / workshop would be more appropriate for this paper.